Microbiology
Spectrum

# The mechanism of action of auranofin analogs in *B. cenocepacia* revealed by chemogenomic profiling

Dustin T. Maydaniuk,[1] Brielle Martens,[1] Sarah Iqbal,[1] Andrew M. Hogan,[1] Neil Lorente Cobo,[1] Anna Motnenko,[1] Dang Truong,[2] Sajani H. Liyanage,[2] Mingdi Yan,[2] Gerd Prehna,[1] Silvia T. Cardona[1,3]

**ABSTRACT** Drug repurposing efforts led to the discovery of bactericidal activity in auranofin, a gold-containing drug used to treat rheumatoid arthritis. Auranofin kills Gram-positive bacteria by inhibiting thioredoxin reductase, an enzyme that scavenges reactive oxygen species (ROS). Despite the presence of thioredoxin reductase in Gram-negative bacteria, auranofin is not always active against them. It is not clear whether the lack of activity in several Gram-negative bacteria is due to the cell envelope barrier or the presence of other ROS protective enzymes such as glutathione reductase (GOR). We previously demonstrated that chemical analogs of auranofin (MS-40 and MS-40S), but not auranofin, are bactericidal against the Gram-negative *Burkholderia cepacia* complex. Here, we explore the targets of auranofin, MS-40, and MS-40S in *Burkholderia cenocepacia* and elucidate the mechanism of action of the auranofin analogs by a genome-wide, randomly barcoded transposon screen (BarSeq). Auranofin and its analogs inhibited the *B. cenocepacia* thioredoxin reductase and induced ROS but did not inhibit the bacterial GOR. Genome-wide, BarSeq analysis of cells exposed to MS-40 and MS-40S compared to the ROS inducers arsenic trioxide, diamide, hydrogen peroxide, and paraquat revealed common and unique mediators of drug susceptibility. Furthermore, deletions of *gshA* and *gshB* that encode enzymes in the glutathione biosynthetic pathway led to increased susceptibility to MS-40 and MS-40S. Overall, our data suggest that the auranofin analogs kill *B. cenocepacia* by inducing ROS through inhibition of thioredoxin reductase and that the glutathione system has a role in protecting *B. cenocepacia* against these ROS-inducing compounds.

**IMPORTANCE** The Burkholderia cepacia complex is a group of multidrug-resistant bacteria that can cause infections in the lungs of people with the autosomal recessive disease, cystic fibrosis. Specifically, the bacterium Burkholderia cenocepacia can cause severe infections, reducing lung function and leading to a devastating type of sepsis, cepacia syndrome. This bacterium currently does not have an accepted antibiotic treatment plan because of the wide range of antibiotic resistance. Here, we further the research on auranofin analogs as antimicrobials by finding the mechanism of action of these potent bactericidal compounds, using a powerful technique called BarSeq, to find the global response of the cell when exposed to an antimicrobial.

**KEYWORDS** Burkholderia, BarSeq, glutathione, ROS, synthetic lethality, antibiotic

Address correspondence to Silvia T. Cardona, silvia.cardona@umanitoba.ca.

Brielle Martens and Sarah Iqbal contributed equally to this article.

The authors declare no conflict of interest.

See the funding table on p. 17.

There is a dire need to develop new antibiotics against the growing threat of antibiotic resistant bacteria. One approach to alleviate this crisis is to repurpose drugs already on the market (1, 2). An example is auranofin, a gold-containing, anti-rheumatic drug found to have bactericidal activity against Gram-positive bacteria (3). Auranofin inhibits thioredoxin reductase (TrxB), disrupting the cell's ability to withstand oxidative stress (3, 4). However, auranofin is inactive in many Gram-negative bacteria

(3, 5). It is not clear whether the lack of activity is due to differences in cell envelope permeability or the presence of glutathione reductase (GOR), which is functionally and structurally similar to TrxB (6).

GOR and TrxB belong to the two major systems that combat reactive oxygen species (ROS) in bacteria, the glutathione and the thioredoxin systems, respectively. The thioredoxin system uses the protein thioredoxin (Trx) and NADPH as a source of electrons (6). Most Gram-negative bacteria encode TrxB and two thioredoxins (6). The thioredoxin protein donates electrons to the ribonucleotide reductase, which creates deoxyribonucleotides from ribonucleotides (7, 8). The oxidized thioredoxin is reduced by TrxB with electrons sourced from NADPH (7, 8). In the glutathione system, glutathione in its reduced form (GSH) can detoxify ROS and reactive nitrogen species (RNS) by reducing ROS/RNS and itself becoming oxidized (GSSG), forming a disulfide bond between two molecules of glutathione (7, 9). Organisms that use glutathione to detoxify ROS/RNS recycle oxidized glutathione to its reduced form by the action of GOR (7, 9). This enzyme uses electrons from NADPH to accomplish this reaction and has a domain for NADPH binding (9, 10).

Bacteria of the *Burkholderia cepacia* complex (Bcc) are a group of Gram-negative opportunistic pathogens (11) that are naturally resistant to many classes of antibiotics and cause severe infections in people with cystic fibrosis (CF) (12, 13). A Bcc member, *Burkholderia cenocepacia* can scavenge ROS using the thioredoxin system, the glutathione system, and various superoxide dismutases and catalases (14, 15). Having these systems allows *B. cenocepacia* to survive intracellularly within macrophages (16), as macrophages use ROS to fight invading bacteria and other pathogens (17). Antibacterial drugs that target ROS protective mechanisms may then be useful in alleviating Bcc infections. To that aim, we tested auranofin and synthesized sugar-modified analogs against Gram-negatives (18). Two of these analogs, MS-40 and MS-40S, exhibited bactericidal activity against *B. cenocepacia*, without detectable resistance (18).

Here, we explore the molecular targets and global mechanisms of action of MS-40 and MS-40S in *B. cenocepacia*. We first identified that the bacterial TrxB is inhibited by auranofin, MS-40, and MS-40S *in vitro*. However, these compounds were inactive against the bacterial GOR. We next found that both MS-40 and MS-40S increased cellular levels of reactive oxygen species, possibly caused by the inhibition of the TrxB. Finally, we exposed a randomly barcoded transposon mutant library (19) to MS-40, MS-40S, and other ROS inducers and quantified relative mutant fitness with BarSeq (20). We found genetic elements related to ROS metabolism that when disrupted caused differential susceptibility to the auranofin analogs, suggesting their involvement in the mechanism of action of the auranofin analogs.

## RESULTS

### MS-40 and MS-40S inhibit TrxB but not GOR in *B. cenocepacia*

Auranofin and other auranofin analogs inhibit *Staphylococcus aureus* and *Mycobacterium tuberculosis* (3) as well as the mammalian TrxB (21). Auranofin is thought to be inactive in Gram-negatives due to the presence of the secondary antioxidant system, glutathione system, or the low permeability of the cell envelope. The *B. cenocepacia* K56-2 genome encodes the *trxB* and *gor* genes and a previous TnSeq analysis identified *trxB* (but not *gor*) as essential in *B. cenocepacia* (22). We hypothesized that MS-40 and MS-40S (Fig. 1A) are active in *B. cenocepacia* because they target TrxB and the structurally and functionally similar protein GOR of the glutathione system, while auranofin is inactive because it only inhibits TrxB.

To confirm that auranofin, MS-40, and MS-40S share the same target, the *trxB* gene from *B. cenocepacia* K56-2 was expressed in *E. coli*, the protein purified (Fig. S1), and the enzymatic activity of TrxB was tested. Auranofin, MS-40, and MS-40S decreased TrxB activity *in vitro* in the nanomolar range (Fig. 1B), indicating that these three compounds are potent inhibitors of this essential protein.

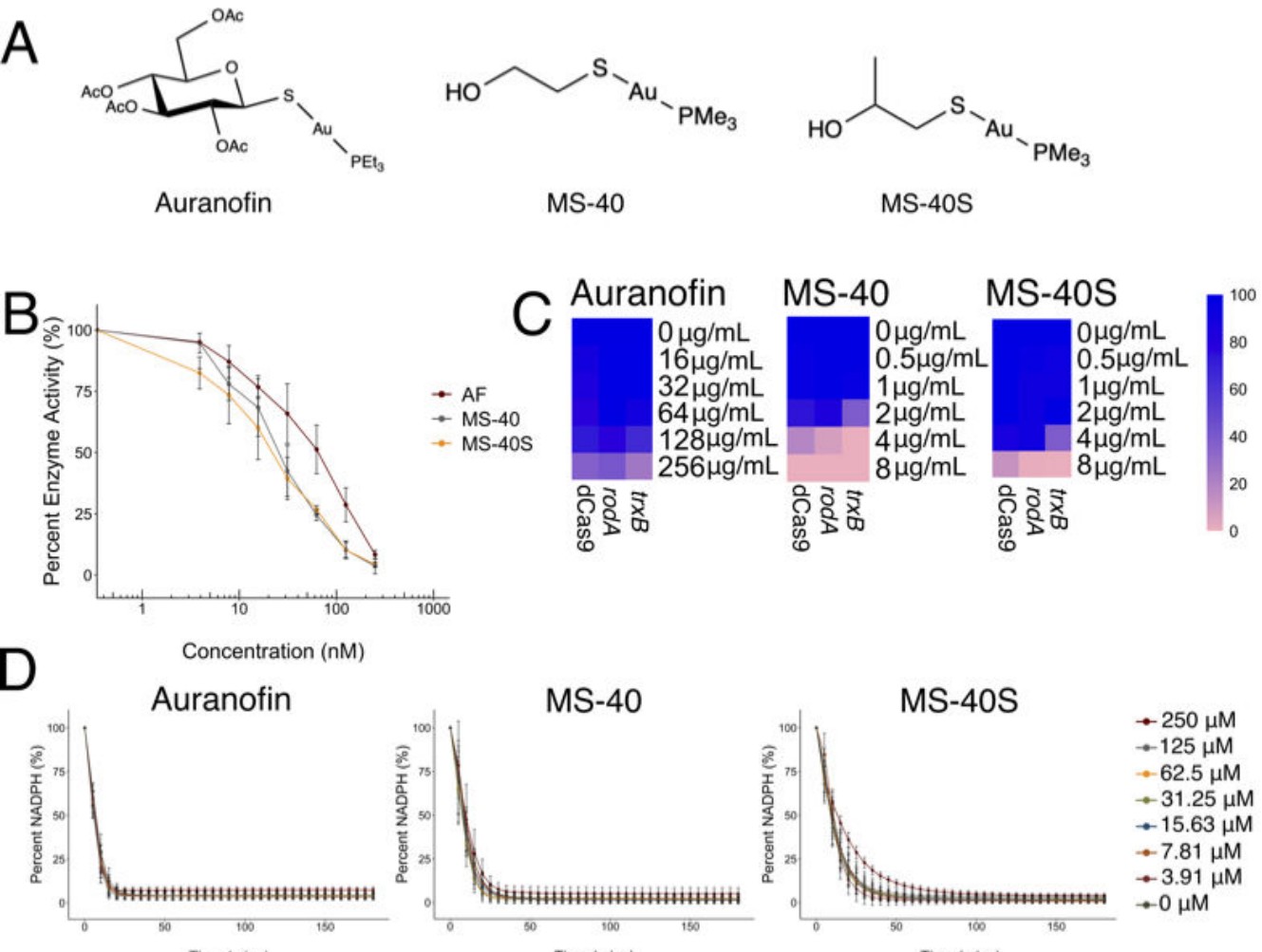

**FIG 1** Auranofin, MS-40, and MS-40S inhibit TrxB but do not inhibit GOR. (A) Chemical structures of auranofin, MS-40, and MS-40S. (B) TrxB *in vitro* assay. Purified TrxB from *B. cenocepacia* K56-2 in the presence of substrate DTNB is reduced to TNB, with NADPH as a source of electrons, producing a fluorescent product. TrxB was exposed to a concentration gradient of auranofin (AF), MS-40, and MS-40S from 250 nM to 3.906 nM. Absorbance at 412 nm was measured in a Synergy-2 plate reader every 5 minutes for 3 hours. Circles represent data from $n = 3$ independent experiments, with the circles showing the mean and error bars representing s.d. (C) CRISPRi mutant of *trxB* gene as well as the dCas9 non-targeting control and *rodA* gene, as a functionally unrelated essential gene control, at MacFarland Standard 0.5 were exposed to a concentration gradient of auranofin, MS-40, and MS-40S. $OD_{600nm}$ was read after 18 hours. Mean of percent growth (relative to no treatment) from $n = 3$ independent experiments are shown, with 100% growth represented as blue and 0% growth represented as pink. (D) Purified GOR from *B. cenocepacia* K56-2 in the presence of substrate, oxidized glutathione is reduced to reduced glutathione, with NADPH as a source of electrons. Absorbance at 340 nm was used to track the usage of NADPH, with readings every 5 minutes for 3 hours on a Synergy-2 plate reader. Circles represent the mean from $n = 3$ independent experiments with error bars showing s.d

Drug target interactions can be also visualized *in vivo* when reduced levels of the target increase the susceptibility of a bacterial strain in the presence of a drug (23). We reasoned that knocking down *trxB* would result in enhanced growth inhibition in the presence of auranofin and auranofin analogs. To address this hypothesis, we investigated the growth phenotype of a *trxB* knockdown mutant exposed to auranofin, MS-40, and MS-40S. The knockdown mutant was created with a CRISPRi system adapted for *Burkholderia* (24), where a rhamnose-inducible dCas9 is expressed, inhibiting *trxB* gene expression. Under gene silencing conditions (0.5% rhamnose), the *trxB* CRISPRi mutant showed slightly enhanced susceptibility to MS-40 and MS-40S (Fig. 1C), which supported the role of TrxB in the mechanism of action. However, the *trxB* CRISPRi mutant did not show as strong hypersusceptibility to auranofin compared to MS-40 and MS-40S,

suggesting auranofin may not reach the cytoplasmic target due to the cell envelope barrier.

Alternatively, auranofin could be inactive in *B. cenocepacia* because GOR can compensate for the lack of TrxB, while auranofin analogs are active because the analogs could also inhibit GOR. To determine whether GOR is a target of MS-40 and MS-40S the *B. cenocepacia* K56-2 *gor* gene was overexpressed and the protein was purified (Fig. S2).

To measure GOR enzymatic activity, we followed the decrease in NADPH, as it is oxidized to NADP when oxidized glutathione is converted to reduced glutathione by GOR (7, 9, 10). The addition of auranofin, MS-40, and MS-40S up to a concentration of 250 µM did not inhibit GOR, indicating that GOR is neither a target of auranofin nor auranofin analogs (Fig. 1D). In the presence of a known GOR inhibitor, 2-AAPA (25), purified GOR from *B. cenocepacia* lost approximately half of its activity at 250 µM (Fig. S3). Thus, GOR is not a target for auranofin, MS-40, or MS-40S and auranofin may be inactive in *B. cenocepacia*, and other Gram-negatives, because of the low permeability of their cellular envelope.

## MS-40 and MS-40S cause an increase in ROS

As we showed that both MS-40 and MS-40S inhibit TrxB *in vitro,* and the thioredoxin system is involved in detoxifying ROS (6), we hypothesized that MS-40 and MS-40S may cause an increase in ROS. To test this, we examined whether *B. cenocepacia* K56-2 had an increase in ROS after exposure to MS-40 and MS-40S using the cell-permeant indicator dye 2′,7′-dichlorodihydrofluorescein diacetate ($H_2$DCFDA). $H_2$DCFDA diffuses into cells in a non-fluorescent form, where it is oxidized by ROS (mainly $H_2O_2$, ROO•, and ONOO⁻) to the fluorescent 2′,7′-dichlorofluorescein (DCF) (26). MS-40 and MS-40S caused a significant increase in ROS compared to the DMSO solvent control (Fig. 2A). Tetracycline, which was used as a control, did not cause an increase in ROS in our conditions, in agreement with a previous report (27). This effect was more pronounced when the concentration of the compounds increased to fourfold the MIC (Fig. 2A). The increase in ROS was also shown using microscopy, in which cells fluoresced only when exposed to MS-40 and MS-40S (Fig. 2B). Together, these results validate that ROS does in fact play a role in the mode of action for the auranofin analogs.

## BarSeq analysis of cells exposed to MS-40, MS-40S, and other ROS inducers

To investigate the full global response of *B. cenocepacia* K56-2 due to exposure to MS-40 and MS-40S, we used randomly barcoded transposon sequencing (BarSeq). A barcoded transposon mutant library in *B. cenocepacia* K56-2 containing more than 330,000 unique mutants in over 6,400 protein-coding genes (19) was exposed to the $IC_{25}$ of MS-40, MS-40S, and the ROS inducers $H_2O_2$, paraquat, $As_2O_3$, and diamide (Table 1) for 8 hours (10–12 generations). $H_2O_2$ and paraquat damage cellular proteins, lipids, and DNA (28–30). Diamide is a thiol stressor that can disrupt the thiol groups in proteins causing disulfide bond formation, potentially disrupting TrxB (31, 32). Lastly, $As_2O_3$ a heavy metal compound binds DNA and also causes an increase in ROS (33, 34). These additional ROS inducers were included as benchmarks for the mechanism of action comparisons. Sequenced barcode read counts served as a proxy for mutant abundance (20) and gene fitness was then calculated as the $log_2$ fold change of a number of reads in experimental condition to time zero (20, 35). The change in gene fitness for each mutant in the presence of a compound compared to the DMSO solvent control was then calculated. A positive gene fitness score indicates the mutant was more abundant following treatment, while a negative fitness score indicates those mutants were more depleted.

We first looked for known interactions with $As_2O_3$, diamide, $H_2O_2$, and paraquat (Fig. 3). One of the most depleted mutants for $H_2O_2$ was one in which *dyp-type peroxidase* (K562_RS21385) was disrupted, which plays a role in ROS scavenging (36). In addition, mutants in known arsenic efflux and arsenic tolerance proteins were identified as being more susceptible to $As_2O_3$ (Fig. 3). These arsenic resistance genes include *arsH*,

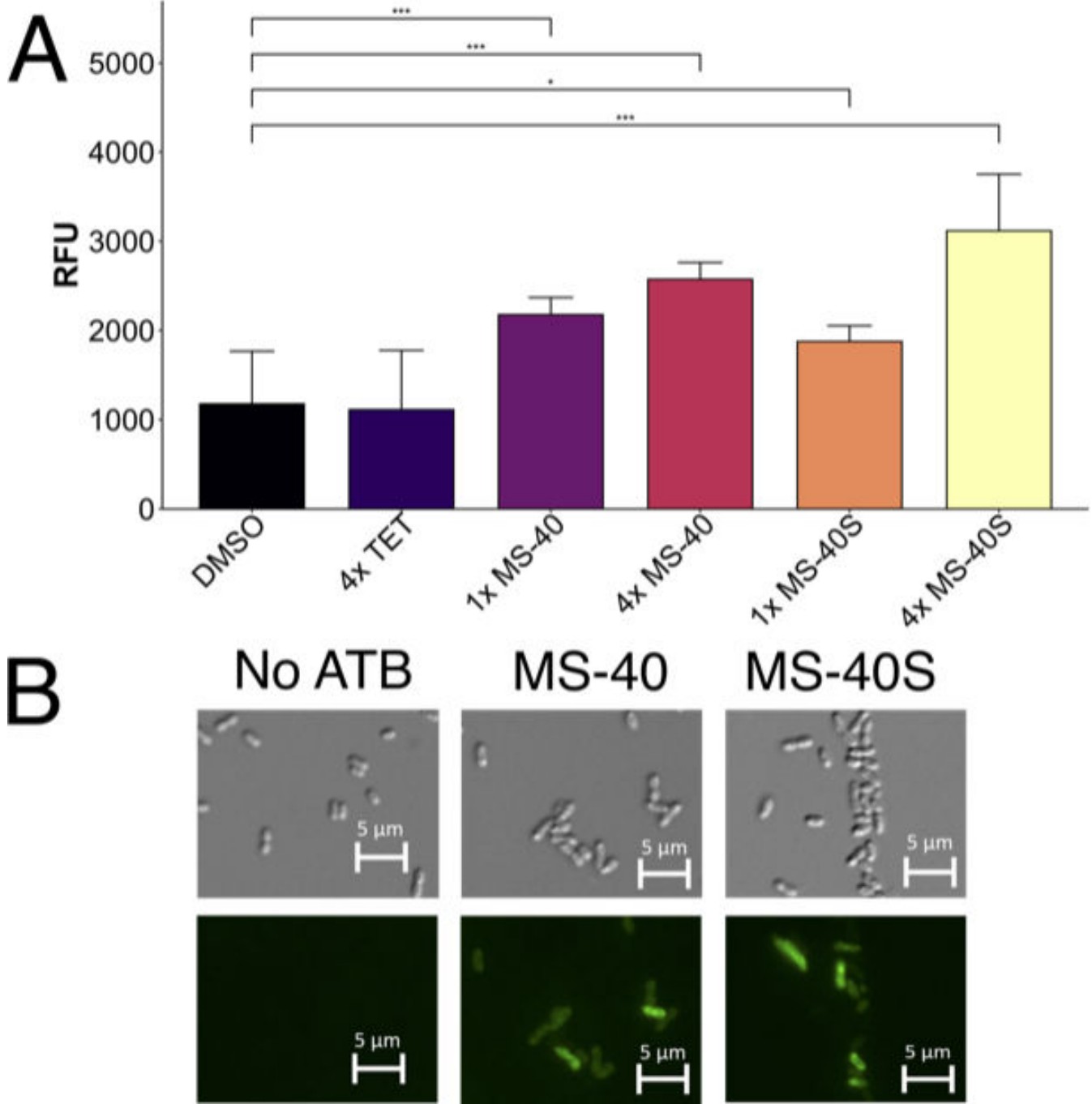

**FIG 2** The MS-40 and MS-40S cause an increase in ROS. (A) ROS detection of MS-40 and MS-40S. Wild-type K56-2 cells, grown to exponential phase, were incubated with $H_2DCFDA$ for 45 minutes at 37°C, then exposed to the corresponding antimicrobial for 3 hours at 1× or 4× their corresponding MICs. Fluorescence ($\lambda_{ex}485/\lambda_{em}528$) was measured on a Synergy-2 plate reader. The mean from $n = 3$ independent experiments are shown in the bar graph, with error bars representing s.d. * indicates $P < 0.05$ and *** indicates $P < 0.0001$ from a one-way ANOVA and post-hoc Dunnett test. (B) Microscopy of wild-type K56-2 cells stained with $H_2DCFDA$ dye then exposed to MS-40 and MS-40S.

which encodes for a flavoprotein that uses NADP+ to oxidize As(III) to As(V), conferring resistance (37), and *acr3 family arsenite efflux transporter* (K562_RS18375) which is involved in the efflux of arsenite from the cell (37). In summary, these results confirm that exposure conditions followed by BarSeq can reliably detect genetic elements that are part of the mechanism of action of the auranofin analogs.

**TABLE 1** Inhibitory concentrations used in BarSeq

| Compound | IC$_{25}$ | Mechanism of action |
|---|---|---|
| MS-40 | 5.0 µM | Unknown; thought to include ROS and thiol homeostasis |
| MS-40S | 8.0 µM | Unknown; thought to include ROS and thiol homeostasis |
| As$_2$O$_3$ | 40.0 µM | Heavy metal, may cause ROS |
| Diamide | 1.30 mM | ROS and thiol stress |
| H$_2$O$_2$ | 0.12 mM | ROS |
| Paraquat | 0.38 mM | ROS |

A genetic network map was generated to visualize similar and unique responses for each of the compounds analyzed, in terms of which genes had a significant change in gene fitness compared to the DMSO control (Fig. 4A). Transposon disruption of 825 genes resulted in significant changes in gene fitness in the presence of MS-40, with 216 exclusive interactions. For MS-40S, we observed significant changes in fitness for 788 genes, with 178 genes exclusive to MS-40S. While unique interactions with MS-40 were related to the glyoxylate shunt, octane oxidation, and general degradation/utilization/

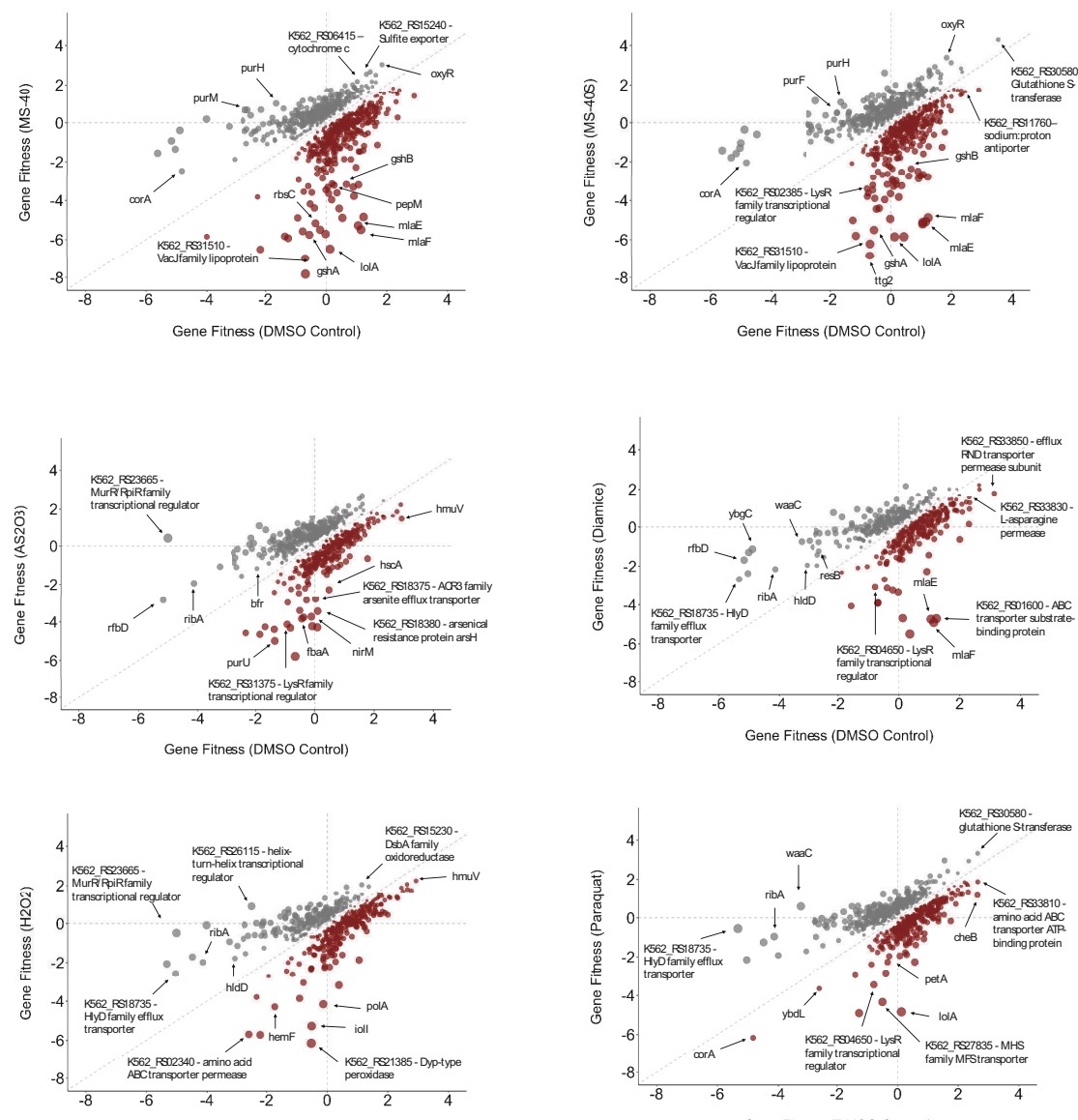

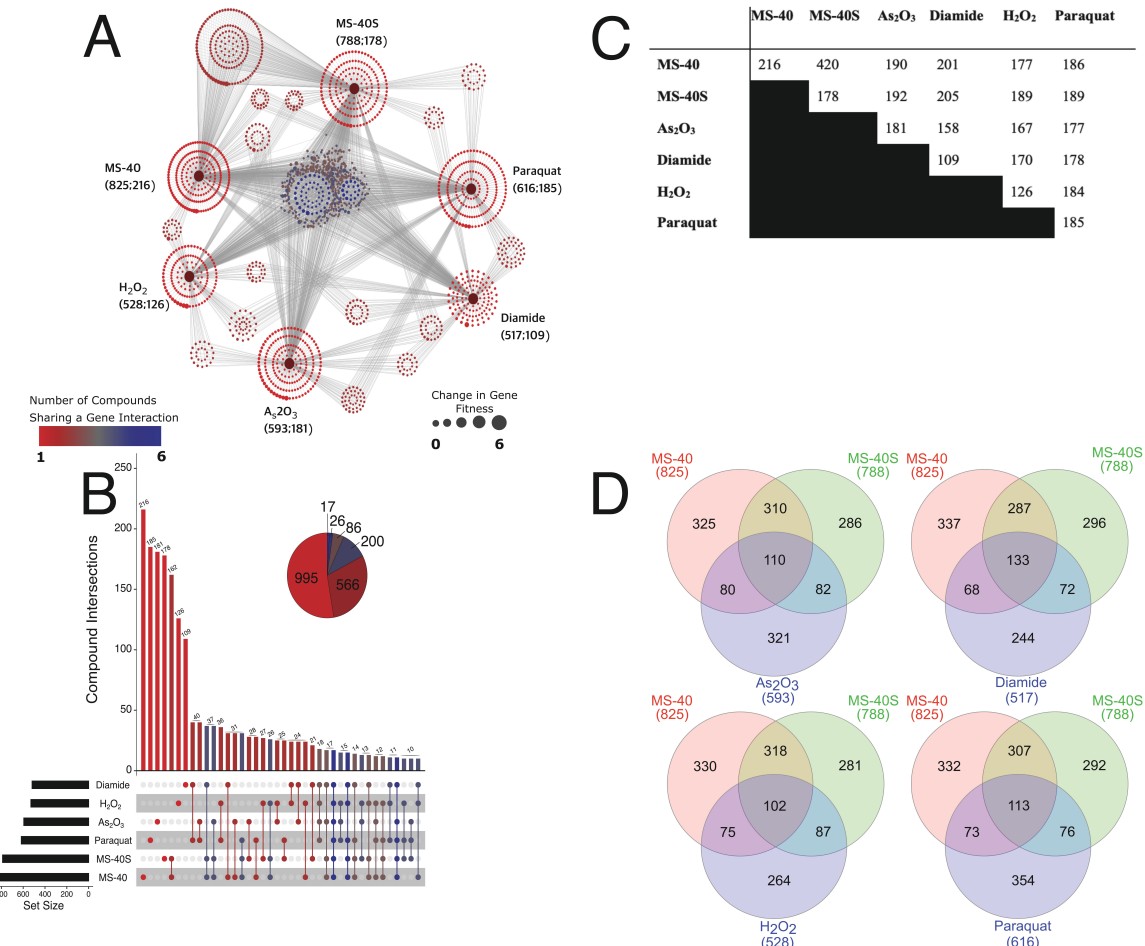

**FIG 4** The MS-40 and MS-40S have similar gene fitness profiles as benchmarking controls. (A) Genetic network map showing gene connectedness between compounds used in the BarSeq figure made using Cytoscape. This figure shows how many genes were identified as significant for each compound and how many were only identified in that condition. Large burgundy dots represent each compound, and gray lines connect each compound to a smaller node that represents a gene, which is colored based on how many compounds that gene, when disrupted, had altered sensitivity. (B) Upset plot showing how many mutants had altered sensitivities in each condition. (C) The number of genes with significant changes in gene fitness between two compounds used in BarSeq. (D) Venn diagrams of genes that had significant changes in gene fitness between MS-40, MS-40S, and a ROS-inducing compound used in the BarSeq. Venn diagrams were generated from Venny 2.1.

assimilation pathways, unique interactions with MS-40S were related to L-glutamate, L-aspartate, and sugar degradation. We observed 593, 517, 528, and 616 genes having significant changes in fitness, with 181, 109, 26, and 185 exclusive interactions for $As_2O_3$, diamide, $H_2O_2$, and paraquat, respectively. Interestingly, the other ROS inducers have notably fewer interactions than MS-40 and MS-40S.

There were 17 genes with differential fitness for all six compounds used in the BarSeq (Fig. 5A and B). Interestingly, one of these genes was *gor* and this gene had positive fitness scores for all compounds, except diamide. The uncharacterized LysR type transcriptional regulator (K562_RS18170) had a negative fitness score for all ROS inducers. In addition, the Upset plot shows how many interactions are shared between multiple conditions (Fig. 4B). Unsurprisingly, MS-40 and MS-40S had the most similar interaction profiles compared to any other compound, with 162 genes shared between these two. Genes in this subset are related to phosphonate/phosphorous metabolism and aerobic respiration. There were 37 interactions shared between just MS-40, MS-40S, and $As_2O_3$, with these genes being related to the NAD salvage pathway, purine nucleotide biosynthesis, and glycerol degradation (Fig. 4B). There were also

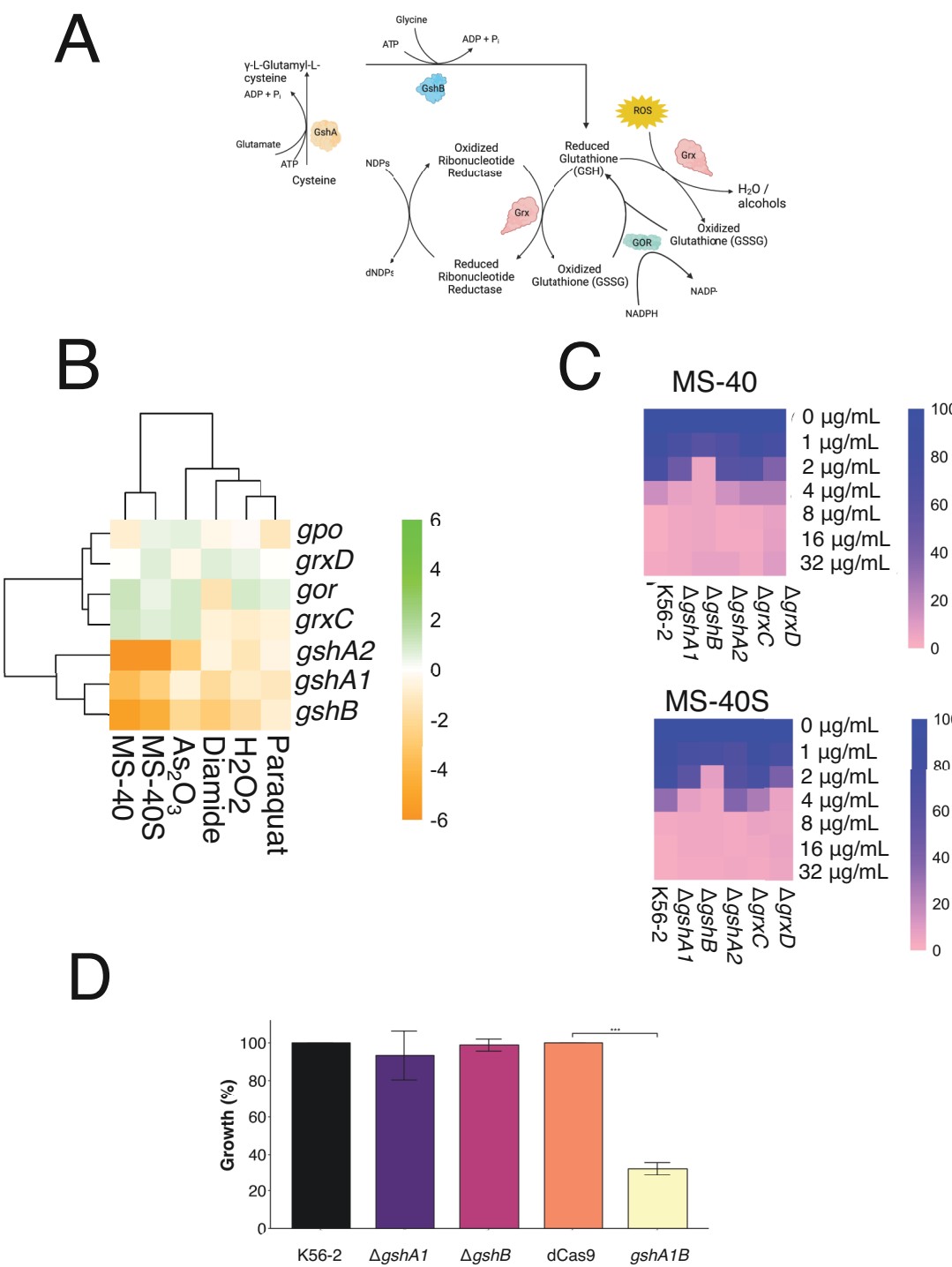

**FIG 5** The glutathione system may be involved in the MOA of MS-40 and MS-40S. (A) Schematic of the glutathione system showing the roles of each protein and their respective reactions. (B) Fitness profile of glutathione-related genes from BarSeq. Heatmap of ROS-related genes and their corresponding gene fitness values for each compound used in BarSeq. Heatmap was generated on RStudio using the pheatmap package. Green represents positive fitness, and orange represents negative fitness. (C) Percent growth (100% growth blue, 0% growth pink) of glutathione-related deletion mutants and wild-type against MS-40 and MS-40S. OD values were taken from MIC testing using broth microdilution assay and converted to percent growth. (D) Growth of each strain was measured 18 hours after grown in LB (for K56-2 wild type, Δ*gshA*, and Δ*gshB*) or LB Tp100 +30.5 mM rhamnose (for dCas9 non-targeting control and *gshA1gshB* CRISPRi mutant). *** indicates $P < 0.0001$ from one-way ANOVA and post hoc Dunnett test.

37 interactions shared between just MS-40, MS-40S, and diamide, with these genes being related to glutathione biosynthesis and L-selenocysteine biosynthesis. For MS-40, MS-40S, and $H_2O_2$, there were 26 shared interactions with the genes being involved in pyrroloquinoline quinone biosynthesis and glutathione biosynthesis. Lastly, there were 31 interactions shared between MS-40, MS-40S, and paraquat, with the genes being involved in flavin biosynthesis and glycerophosphodiester degradation.

A pairwise comparison of compound response shows that MS-40 and MS-40S seem to share a similar number of interactions with each of the benchmarking compounds (Fig. 4C). MS-40 and MS-40S shared 110 genes with $As_2O_3$, 133 with diamide, 102 with $H_2O_2$, and 113 with paraquat (Fig. 4D). Together, these results suggest that MS-40 and MS-40S share some of the mechanism of action with the tested ROS inducers.

## BioCyc analysis identifies glutathione and purine biosynthesis in the mechanism of action for MS-40 and MS-40S

As BarSeq captured the *B. cenocepacia* global response to ROS inducers, we set to identify related pathways that were important for MS-40 and MS-40S's mechanism of action. To help narrow down which cellular processes highly affect each compound used in BarSeq, analysis from BioCyc was used (38). Pathway enrichment (Fig. S4) was used to find whether the BarSeq data were enriched for mutants in specific pathways. BioCyc analysis finds which pathways were enriched or depleted in the set of genes that had a significant change in gene fitness as compared to the entire genome of *B. cenocepacia* K56-2, without taking the magnitude of the gene fitness value into consideration. For genes that had a positive fitness (Fig. S4A), the gene set was especially enriched in genes related to nucleoside and nucleotide metabolism, specifically purine metabolism, including 5-aminoimidazole ribonucleotide biosynthesis. Diamide and paraquat had pyrimidine biosynthesis as an enriched pathway. Alternatively, for genes that had a negative fitness for MS-40 and MS-40S, the gene set was enriched in genes in amine and polyamine degradation, glutathione biosynthesis, and phosphorus compound metabolism (Fig. S4B).

BioCyc also analyzes a data set for a pathway perturbation score (PPS), which is calculated as the weighted average of each gene's fitness belonging to a pathway (38). For genes that had a positive gene fitness, the top perturbed pathways for both MS-40 and MS-40S were the two-component alkanesulfonate monooxygenase (Fig. S5A), which is involved in using alkanesulfonates as a source of sulfur (39). Other perturbed pathways in this analysis were related to purine metabolism, such as biosynthesis and salvage. In addition, other ROS inducers used in the BarSeq also had purine or pyrimidine synthesis/salvage as their most perturbed pathways. Moreover, pyruvate decarboxylation to acetyl-CoA was a pathway that was in the top perturbed pathways for MS-40S and diamide, suggesting a role of central metabolism in the mechanism of action of these compounds. On the other hand, for genes that had a negative gene fitness, the top 15 perturbed pathways are shown in Fig. S5B. Glutathione biosynthesis was one of the most perturbed pathways for MS-40, MS-40S, $As_2O_3$, and diamide, suggesting that this pathway is highly responsible for scavenging ROS in the cell exposed to oxidative stress. In addition, purine metabolism was a process that was found to cause decreased susceptibility to all ROS inducers, a novel mechanism in ROS susceptibility. The cellular processes that are putatively involved in the mechanism of action of MS-40 and MS-40S are marked with an asterisk in Fig. S4 and Fig. S5.

As MS-40 and MS-40S caused an increase in ROS (Fig. 2), we looked at mutants in ROS-related genes and their corresponding fitness values. Figure S6 shows which ROS scavenging genes were identified by BarSeq. Some of these genes when disrupted lead to increased susceptibility to MS-40 and MS-40S. These include *katG* and *katE,* encoding catalases organic hydroperoxide resistance (*ohr*) family peroxidase (K562_RS31135), *sodC* (encoding a superoxide dismutase), *ahpC* (encoding an alkyl hydroperoxide reductase), and various genes encoding peroxidases. However, mutants of a putative ROS regulator, *oxyR*, as well as *ahpD* and *ahpF*, which in *E. coli* are known to help clear ROS (29, 40),

showed decreased susceptibility to MS-40 and MS-40S. The other compounds used in the BarSeq experiment also had ROS-related genes having significant changes in gene fitness, in addition to displaying similar susceptibilities as MS-40 and MS-40S (Fig. S6). Based on hierarchical clustering, the response to MS-40 and MS-40S is most similar to $H_2O_2$, suggesting that hydrogen peroxide is the type of ROS produced by MS-40 and MS-40S. It is similar to diamide, which causes thiol stress, suggesting that MS-40 and MS-40S may also cause thiol stress, which might lead to an accumulation of hydrogen peroxide in the cell.

## The glutathione biosynthetic pathway is involved in protection against MS-40 and MS-40S

The putative glutathione biosynthesis pathway of *B. cenocepacia* based on sequence similarity to *E. coli* is shown in Fig. 5A. The role of this system is to create glutathione, which can be used to detoxify ROS inside the cell (9, 10). Glutathione is made from a two-step reaction using glycine, cysteine, and glutamate, with GshA and GshB catalyzing the first and second steps, respectively. The genes *gshA* and *gshB* encode the glutathione synthetic enzymes, *grxC* and *grxD* encode glutaredoxins, and *gpo* encodes a glutathione peroxidase (10).

Our BarSeq analysis shows that transposon disruptions of *gshA1*, *gshA2*, and *gshB* resulted in gene fitness between −4 and −6 in the presence of MS-40 and MS-40S, indicating increased susceptibility (Fig. 5B). Disruption of *gshA1*, *gshA2*, and *gshB* also increased susceptibility to the other ROS-inducing compounds used in the BarSeq experiment. However, the interactions were not as strong as with MS-40 and MS-40S. Thus, we deleted these three genes in *B. cenocepacia* K56-2, as well as two glutaredoxins, *grxC* and *grxD*, and exposed the mutants to MS-40 and MS-40S. The mutants Δ*gshA1* and Δ*gshB* were more susceptible to MS-40 and MS-40S, compared to the wild type (Fig. 5C). We observed that the MIC drops twofold and fourfold for Δ*gshA1* and Δ*gshB*, respectively. These both validate the BarSeq method and indicate the involvement of the glutathione system in the mechanism of action of MS-40 and MS-40S.

The genes *gshA1* and *gshB* form a gene cluster, and efforts to delete the putative operon were unsuccessful, suggesting a synthetic lethality effect. Thus, to investigate the effect of silencing both genes in *B. cenocepacia* K56-2, we created a CRISPRi knockdown which is polar when genes are arranged in an operon. The growth of this CRISPRi mutant was measured together with Δ*gshA1* and Δ*gshB* (Fig. 5D). Neither Δ*gshA1* nor Δ*gshB* had a growth defect compared to wild-type K56-2 but the *gshA1B* CRISPRi mutant had a major growth defect in the presence of rhamnose and showed elevated ROS levels (Fig. S7). We observed a percent growth of 32.5% compared to the non-targeting control strain (Fig. 5D). Thus, *gshA* and *gshB* form a synthetic lethality pair. While silencing *gshA* and *gshB* simultaneously did not change the MIC and minimum bactericidal concentrations (MBC) of MS-40 and MS-40S, it dropped the MIC of ceftazidime from 32 µg/mL to 4 µg/mL and decreased the MBC for ciprofloxacin, ceftazidime, and meropenem between twofold and fourfold (Table 2).

## DISCUSSION

Auranofin and previously published auranofin analogs inhibit TrxB, an enzyme that plays a role in thiol-homeostasis in the cell (3, 21). However, it was unknown whether the auranofin analogs MS-40 and MS-40S share the same target with auranofin or whether they have any other additional targets. It is assumed that the active component of auranofin is the gold atom, which binds the sulfur in the active site of TrxB. This binding inhibits the formation of a critical disulfide bond required for electron transfer between substrates, disrupting the function of the enzyme (21, 41).

We have shown that MS-40 and MS-40S inhibit TrxB, which is expected as auranofin inhibits this enzyme (3, 21). We hypothesized that MS-40 and MS-40S would also inhibit the functionally similar protein GOR, as both contain two key cysteine amino acids in their active site (6, 7). However, this was not the case as MS-40 and MS-40S did not

**TABLE 2** Minimum inhibitory concentration and minimum bactericidal concentration of *gshA1B* CRISPRi mutant against bactericidal antibiotics

| | MIC (µg/mL) | | MBC (µg/mL) | |
|---|---|---|---|---|
| Strain | dCas9[a] | gshA1B | dCas9[a] | gshA1B |
| Ciprofloxacin | 2 | 2 | 16 | 4 |
| Doxycycline | 4 | 4 | >16 | >16 |
| Meropenem | 32 | 32 | 64 | 32 |
| Ceftazidime | 32 | 4 | 64 | 32 |
| MS-40 | 8 | 8 | 16 | 16 |
| MS-40S | 8 | 8 | 16 | 16 |

[a]Non-targeting control. MIC and minimum bactericidal concentrations (MBC) were measured in cation-adjusted Mueller Hinton broth including 100 µg/mL trimethoprim and 0.5% rhamnose.

affect the activity of GOR. In addition, the BarSeq results showed a slightly decreased susceptibility of *gor* mutants to MS-40, MS-40S, and other ROS inducers (Fig. S6). *E. coli* mutants lacking GOR have decreased levels of glutathione (42) but the ratio of reduced to oxidized glutathione remains unperturbed, which suggests GOR has a minor role in ROS protection. As MS40, MS40S, and auranofin inhibit TrxB *in vitro* but only MS40 and MS40S are active *in vivo*, differential restrictions imposed by the cell envelope might be one reason why auranofin is not active in *B. cenocepacia*. This aspect was not further investigated in the current study.

To determine the factors involved in susceptibility and tolerance to two antimicrobials, MS-40 and MS-40S, as well as other ROS-producing compounds as benchmarking controls, we used a randomly barcoded transposon mutant library in *B. cenocepacia* K56-2 previously created in our laboratory (19). The BarSeq approach allowed us to have a genome-wide look at which mutants had altered susceptibilities after exposure to MS-40, MS-40S, and other ROS inducers. In addition, BarSeq facilitated a genome-wide comparison between MS-40, MS-40S, and other ROS inducers. We found the chemical interactions of MS-40 and MS-40S are most similar to diamide and $As_2O_3$, suggesting that the mechanism of action of the auranofin analogs involves thiol stress and response to heavy metal toxicity.

There are several mutants shared between MS-40 and MS-40S in which the disrupted genes result in altering mutant susceptibility. These include *gshA* and *gshB*, which are involved in glutathione biosynthesis, *oxyR*, a regulatory for oxidative stress, and *purH*, *purL*, and *purM* in purine biosynthesis. Similar chemical genetic interaction profiles are expected as MS-40 and MS-40S are structurally very similar. The involvement of these genetic elements suggests that the mechanism of action of MS-40 and MS-40S includes the glutathione system, which can detoxify ROS and maintain thiol homeostasis (9) and oxidative stress in the cell. This is also shown in the pathway perturbation score and pathway enrichment analysis, which again pointed towards glutathione biosynthesis and purine metabolism/biosynthesis.

We further explore the role of the glutathione biosynthesis pathway in *B. cenocepacia* by constructing gene deletion mutants in *gshA1*, *gshA2*, and *gshB*, as well as in two glutaredoxins, *grxC* and *grxD*. Deleting some of these genes in *B. cenocepacia* K56-2 led to a slightly increased sensitivity to MS-40 and MS-40S. In general, we found that pooled growth (as in the BarSeq experiment) enhances susceptibility related to clonal growth, probably due to the additional effect of mutant competition (43).

For ROS-related genes grouped by hierarchical clustering, MS-40 and MS-40S had more similar clustering to $H_2O_2$ and diamide, which might suggest that the ROS produced by MS-40 and MS-40S is $H_2O_2$ and may also cause thiol-stress. In addition, disrupting *oxyR* led to increased tolerance to MS-40 and MS-40S in the BarSeq data and deleting this gene validated this interaction. Namely, the deletion mutant was more tolerant to MS-40, MS-40S, and $H_2O_2$. This is the opposite of what is expected because *oxyR* is an activator of ROS-scavenging genes in many bacteria (29, 40). However, some data show that OxyR can also act as a negative repressor in some bacteria (40). It has

been shown that *oxyR* transcription increases upon exposure to ROS in *B. cenocepacia* (44) but whether it is a positive or negative regulator is not known.

The genes *gshA1* and *gshB* are in a putative operon in *B. cenocepacia* K56-2 and our attempts to delete the gene cluster were unsuccessful. Only silencing both genes resulted in any growth defect but deleting either of the genes alone did not produce a growth defect. We thus created a CRISPRi mutant of this operon in which we could tune the level of suppression, silencing both *gshA1* and *gshB*. CRISPRi is a method for gene silencing and is polar, such that if we silence the first gene in an operon the subsequent genes are also silenced. As the CRISPRi mutant showed a severe growth defect, *gshA1* and *gshB* appear to be a new synthetic lethal combination. We suggest these proteins have at least semi-redundant functions in synthesizing glutathione and thus deleting both may completely abolish glutathione biosynthesis.

## MATERIALS AND METHODS

### Bacterial strains and growth conditions

The plasmids and strains used in this study are listed in Table S1 and S2, respectively. The following selective antibiotics were used for mutant construction: trimethoprim (Sigma; 50 µg/mL for *E. coli* and 100 µg/mL for *B. cenocepacia*), gentamicin (Fisher Scientific; 50 µg/mL for *B. cenocepacia*), kanamycin (Fisher Scientific; 40 µg/mL for *E. coli*), and tetracycline (Sigma; 20 µg/mL for *E. coli* and 100 µg/mL for *B. cenocepacia*). The strains and mutants used in this study are listed in Table S2. *B. cenocepacia* cultures were grown in LB-Lennox (LB) broth at 37 °C with shaking at 230 rpm.

### Auranofin derivatives and antibiotics

Stock solutions at 20 mg/mL of auranofin, MS-40, and MS-40S were prepared by dissolving the compounds in dimethyl sulfoxide (DMSO). Antibiotics were prepared at the following concentrations: ciprofloxacin (Sigma), 10 mg/mL in 0.1 M HCl; tetracycline (Sigma), 10 mg/mL in ethanol; meropenem (Sigma), 10 mg/mL in DMSO; ceftazidime (Sigma), 10 mg/mL in 0.1 M NaOH.

### CRISPRi mutants

CRISPRi mutants were made as previously described (24). Inverse PCR was used to introduce new sgRNA targeting regions into pSCB2-sgRNA. The 5′-GTTTTAGAGCTAGAA ATAGCAAGTTAAAATAAGGC-3′ primer was used with a 5′ extension of the 20-nucleotide targeting sequence. This primer was used with primer 1092 (Table S3) for PCR amplification. The PCR product was used for blunt end ligation using T4 DNA ligase (NEB), T4 PNK (NEB), and *Dpn*I (NEB) in a custom ligase buffer (132 mM Tris-HCl pH 7.5, 20 mM MgCl$_2$, 2 mM ATP, 15% PEG6000 in de-ionized water). The reaction was then incubated at 37°C for 30 minutes. The ligation mix was then transformed into *E. coli* DH5α and tri-parental mating was performed to transfer the plasmid into *B. cenocepacia* K56-2::dCas9.CRISPRi mutants were made as previously described (24). Inverse PCR was used to introduce new sgRNA targeting regions into pSCB2-sgRNA. The 5′-GTTTTAGAGCTAGAAATAGCAAG TTAAAATAAGGC-3′ primer was used with a 5′ extension of the 20-nucleotide targeting sequence. This primer was used with 1092 for PCR amplification. The PCR product was used for blunt end ligation using T4 DNA ligase (NEB), T4 PNK (NEB), and *Dpn*I (NEB) in a custom ligase buffer (132 mM Tris-HCl pH 7.5, 20 mM MgCl$_2$, 2 mM ATP, 15% PEG6000 in de-ionized water). The reaction was then incubated at 37°C for 30 minutes. The ligation mix was then transformed into *E. coli* DH5α and tri-parental mating was done to transfer the plasmid into *B. cenocepacia* K56-2::dCas9.

## Antibiotic susceptibility testing

Antibiotic susceptibility testing or minimum inhibitory concentration experiments was done as described in the CLSI guidelines (45). The compound to be tested was serially diluted twofold in cation-adjusted Mueller Hinton broth (CAMHB). The bacterial strain to be tested was diluted to a turbidity equal to the McFarland standard 0.5 in phosphate-buffered saline (PBS) and then diluted 100-fold in CAMHB, which was then added to the twofold concentration gradient of the compound. For CRISPRi mutants, overnights were grown in LB Tp100 with 0.5% rhamnose and diluted in CAMHB with Tp100 and 0.5% rhamnose. Plates were incubated at 37°C without shaking and the minimal inhibitory concentration (MIC) was read after 20 hours of growth.

## TrxB purification and inhibition assay

The TrxB gene (locus tag; K562_RS14600) was amplified from *B. cenocepacia* K56-2 genome using primers 2303 and 2304 (Table S3). The primers contained the cut sites *BamH*I and *Hind*III to be cloned into pET24b+, which contains a C-terminal 6× His-tag, creating pET24b-TrxB. The plasmid was then transformed into *E. coli* BL21(DE3) cells. For purification, *E. coli* containing pET24b-TrxB was grown in Lysogeny Broth (LB) culture medium to an $OD_{600nm}$ of 0.6 at 37°C for approximately 3 hours with shaking. After that, 1 mM IPTG was added to induce expression of *trxB*, and the cells were allowed to grow for an additional 3 hours at 30°C with shaking. Cells were pelleted *via* centrifugation at $4,100 \times g$ and resuspended in lysis buffer: 50 mM Tris pH 7.5, 500 mM NaCl, 25 mM imidazole and lysed using an Emulsiflex-C3 High Pressure Homogenizer (Avestin). The cells were centrifuged at $35,000 \times g$, lysed by sonication and the pellet and supernatant were separated by centrifugation and visualized by Coomassie-stained SDS-PAGE gel (Fig. S2). Protein was then purified by affinity chromatography using a nickel NTA-agarose column (Bio-Rad). Size exclusion was performed using an AKTApure (Cytiva) with an SD75 16/600 column. The buffer used for size exclusion was 20 mM Tris pH 7.5, 150 mM sodium chloride, and 1 mM β-ME. To confirm protein purity from chromatography, samples were run on a 10% SDS-PAGE gel and visualized by Coomassie-stained SDS-PAGE gel (Fig. S2) The *In vitro* inhibition assay was done as described in the protocol from the Sigma kit (catalog number CS0170). Manufacturer instructions were followed except for the TrxB protein, which we substituted with our purified protein. The undiluted enzyme was diluted 20-fold in assay buffer, then 2 µL was added to each well, thus the resulting concentration was 0.0091 units/mL. Inhibition of TrxB was measured by the reaction of 5,5′-Dithiobis(2-nitrobenzoic) acid (DTNB) to 5-thio-2-nitrobenzoic acid (TNB), catalyzed by TrxB, which is detectable at 412 nm. MS-40 and MS-40S were serially diluted in concentrations ranging from 250 nM to 3.906 nM. $A_{412nm}$ was measured using a BioTek Synergy 2 plate reader. Activity was reported in percent enzymatic activity measured at 412 nm absorbance relative to 0 nM enzyme concentration negative control ($OD_{412\ nm}$ at 3 hours for experimental condition/$OD_{412\ nm}$ at 3 hours for 0 nM control).

## GOR purification and inhibition assay

GOR (locus tag K562_RS03010) was amplified from the *B. cenocepacia* K56-2 genome using the primers 3002 and 3385 (Table S3). Primers added *Nde*I and *Hind*III cut sites for cloning into pET-22+, containing a C-terminal 6× His-tag, creating pET-22-GOR. This plasmid was then transformed into *E. coli* BL21 (DE3)-Gold cells and GOR was purified as stated with TrxB (Fig. S3). *In vitro* inhibition assay was done as described in the protocol from the Sigma kit (catalog number GRSA). Manufacturer instructions were followed except for the GOR protein, which we substituted with the *B. cenocepacia* K56-2 GOR. The purified protein was diluted 20-fold in assay buffer, then 2 µL was added to each well, thus the resulting concentration was 0.0052 units/mL. The activity of GOR was measured by tracking the presence of NADPH in the following reaction: $NADPH + H^+ + GSSG \rightarrow 2GSH + NADP^+$ which is catalyzed by GOR. Enzyme activity was determined by tracking the absorbance at 340 nm as an indicator of NADPH levels (46). MS-40, MS-40S, Auranofin,

and R,R′−2-Acetylamino-3-[4-(2-acetylamino-2-carboxyethylsulfanylthiocarbonylamino)-phenylthiocarbamoylsulfanyl]propionic acid, hydrate (2-AAPA) were serially diluted in concentrations ranging from 250 µM to 3.906 µM.

## ROS detection

This protocol was adapted from references (47) and (48). An overnight culture of *B. cenocepacia* K56-2 was subcultured in 5 mL LB and grown for 3 hours, then adjusted to an $OD_{600nm}$ of 0.3 in 8 mL 1× PBS (pH 7.4). The dye 2′,7′-dichlorodihydrofluorescein diacetate (H2DCFDA) (Sigma) at 10 µM was added to the culture and incubated at 37℃ with shaking for 45 minutes, protected from light. After incubation, cultures were washed at 5,000 rcf for 5 minutes and resuspended in PBS. Cultures were combined with 4× MIC solutions of MS-40, MS-40S, and tetracycline and 1× MIC of MS-40 and MS-40S. Each condition was added to a 96-well plate, along with exponential phase cells with and without dye, and a DMSO control, all of which were added in triplicate. Empty wells were filled with LB to prevent evaporation. Fluorescence with excitation at 485 nm and emission at 535 nm were measured every 30 minutes for 14 hours using a BioTek Synergy 2 plate reader. One-way ANOVA and Dunnett test were performed using R studio.

CRISPRi mutants were grown overnight in LB with 100 µg/mL trimethoprim and 0.5% rhamnose. The cells were subcultured in the same conditions for 3 hours to reach the exponential phase. Once in the exponential phase, H2DCFDA dye was added and allowed to incubate for 45 minutes at 37℃. The cells were then washed twice and resuspended in PBS with Tp100 and 0.5% rhamnose. Non-targeting dCas9 CRISPRi mutant was used as a control. Fluorescence (λ excitation = 485 nm, λ emission 528 nm) and $OD_{600nm}$ readings were taken every 5 minutes for 2 hours in a BioTek Synergy 2 multimode plate reader. The ratio of ROS production was calculated by calculating relative fluorescence units (RFU)/OD, this was then normalized to the rate of ROS production of the control. Autofluorescence of H2DCFDA dye with no cells in PBS was considered in calculations. One-way ANOVA and Dunnett test were performed using R studio.

## Microscopy

*B. cenocepacia* K56-2 was grown in LB until stationary phase then back diluted to an approximate $OD_{600nm}$ of 0.025 in LB and allowed to grow to mid-exponential phase, about 3 hours. Then the cells were exposed to the H2DCFDA dye for 45 minutes, washed with PBS then exposed to the antibiotics for 3 hours at the respective concentrations. Next, 1 mL of cells was centrifuged and washed with PBS. Cells in PBS were spotted on 1.5% agarose pads and imaged by DIC microscopy on an Axio Imager upright microscope (Carl Zeiss Microscopy GmbH). Fluorescent microscopy was also done with the Axio Imager upright microscope with λ excitation = 450–490 nm, λ emission 500–550 nm.

## Barcoded transposon mutant pool exposure, library preparation, and sequencing

The BarSeq exposure, library preparation, and sequencing were done as previously described (19). The library was thawed and mixed to have equal an abundance of each of the unique mutants. This pool was then inoculated at an $OD_{600nm}$ of 0.025 in 50 mL LB with 0.2% rhamnose, which was then grown at 37 ℃ with shaking at 230 rpm. This culture was grown to an early exponential phase, an $OD_{600nm}$ of approximately 0.15. This culture was then aliquoted into 2 mL test tubes. A 2 mL tube was used as time zero and not exposed to any antibiotics. The other remaining cultures were exposed to the $IC_{25}$ of each compound tested and grown for 8 hours, along with a 1% DMSO solvent control. These cultures were grown at 37℃ with shaking at 230 rpm.

Genomic DNA was extracted using the PureLink Genomic DNA Mini Kit (Invitrogen) after compound exposure. To amplify the barcodes, we added 200 ng genomic DNA,

20 µM of primer and each product from each condition was amplified with a unique i7 and i5 index, and Q5 high-fidelity DNA polymerase with the high-GC buffer, and standard Q5 reaction buffer (NEB). To purify the 196 bp desired product, two rounds of dual size selection with Sera-Mag Select (Cytiva) magnetic beads were used with 200 µL of BarSeq PCR products. A NextSeq 500 in high-output mode (Donnelly Centre, Toronto, Canada) was performed with reagent kit v2.5 and 20% PhiX spike. This generated 410–510 million 30 bp single-end reads. The primers used can be found in Table S3.

## Fitness calculation for BarSeq

The scripts from reference (20) were used to associate each BarSeq reads with the correct barcode. These reads, in fastq format, were trimmed using the FASTX toolkit (http://hannonlab.cshl.edu/fastx_toolkit/) to contain only the 20-nucleotide barcode, with reads containing a quality score less than 20 being filtered out. NNNGTCGACCTGCAGCGTACG and AGAGACC are artificial pre- and post-sequences, respectively, that were added to all barcodes to make them valid according to the RunBarSeq.pl script.

Next, the scripts from reference (49) were used to process the barcodes to compare the abundance of each mutant between conditions (https://github.com/Dutton-Lab/RB-TnSeq-Microbial-interactions). With log transformations, 0 values create errors, thus, a pseudocount of 0.1 was added to all barcode counts. Barcodes were removed if they (1) had fewer than three reads in the Time 0 condition and (2) represented intergenic insertions, or insertions in the first or last 10% of a gene. Normalization of raw mean read counts was done against 10 non-essential genes that showed no fitness effect in any condition. These were: K562_RS24650 (ABC-type amino acid transporter), K562_RS22855 (putative glycosyltransferase), K562_RS05000 (hydrolase family protein), K562_RS12100 (acyl-CoA dehydrogenase), K562_RS01045 (Raf kinase inhibitor-like protein), K562_RS06455 (putative PHA depolymerase protein), K562_RS13470 (*gudD*, glucarate dehydratase), K562_RS16220 (DUF3025 domain-containing protein), K562_RS18550 (hypothetical protein), and K562_RS28510 (hypothetical protein). Fitness values for each strain were calculated as the $\log_2$(reads in experimental condition/reads in Time 0). The fitness value for each gene was calculated as the arithmetic mean from the individual strain fitness values for each gene. Smoothing of fitness values was done using a genomic position using a moving window to deal with increased read count due to proximity to the replication forks (20). Gene fitness for the three replicates was calculated as the inverse-variance weighted mean. Spearman, Pearson, and Lin correlation coefficients across replicates for each condition were between 0.5 and 0.8 and are shown in Table S4. An independent two-sided Student's *t*-test was performed on each gene fitness value compared to the DMSO control for each condition.

## BioCyc pathway enrichment and pathway perturbation score

For BioCyc analysis (38), a text file containing locus tags for genes that had significant changes in gene fitness, along with the said gene fitness value, was uploaded for each compound into a SmartTable. This was done separately for genes that had negative fitness scores and positive fitness scores. To calculate the pathway perturbation score, the cellular overview tool was used, and, under Overlay Experimental Data, the data from SmartTable were imported, and the pathway perturbation score was calculated using the default settings. For pathway enrichment, in the SmartTable, the enrichment tool for pathways was used with the default settings.

## Unmarked gene deletions

A double homologous recombination method described previously (50, 51) was used to make unmarked gene deletions in *B. cenocepacia* K56-2. Approximately 350 bp of upstream and downstream regions were amplified and ligated using a *Hind*III restriction cut site. This ligated product was used to amplify the upstream and downstream regions fused, which was then ligated into the pGPI-SceI plasmid using *Xba*I and *Xma*I cut

sites. These plasmids were then transformed into *E. coli* SY327 and conjugated into *B. cenocepacia* K56-2. Colony PCR was used to confirm the insertion of the first recombination event of trimethoprim-resistant clones. To encourage the second recombination event, pDAI-SceI-*sacB* was conjugated into these clones. Trimethoprim-sensitive colonies were identified and colony PCR was used to confirm the deletion. The pDAI-SceI-*sacB* plasmid was cured by patching onto LB (without NaCl) +15% sucrose. Colonies that grew on sucrose were screened for tetracycline sensitivity for successful curing of the plasmid. Deletion was confirmed using colony PCR. Colony PCR primers can be found in Table S3. Gel-confirming deletions can be found in Fig. S8. Primers were designed upstream (forward primer) and downstream (reverse primer) for each gene outside of the homologous regions, to determine whether the corresponding region was deleted, as seen as a decrease in colony PCR band size. Colonies were resuspended in 50 µL sterile water and used as templates. Q5 high-fidelity DNA polymerase with the high-GC buffer, and standard Q5 reaction buffer (NEB) was used.

## Growth detection of glutathione mutants

To measure the growth of the glutathione mutants, cultures were made in LB for wildtype and deletion mutants or in LB plus 100 µg/mL trimethoprim for the non-targeting strain or the *gshA1gshB* CRISPRi mutant. Once they reached the stationary phase, each strain was back diluted to an $OD_{600nm}$ of 0.01 in LB or LB Tp100 ± 0.5% rhamnose in a 96-well plate. The plate was incubated at 37°C with shaking for 18 hours. $OD_{600nm}$ was then measured to determine the growth of each mutant. One-way ANOVA and Dunnett test were performed using R studio.

## ACKNOWLEDGMENTS

This work was supported financially by grants from the Canadian Institutes of Health Research (CIHR) grant and Natural Sciences and Engineering Research Council of Canada (NSERC) awarded to STC and a Natural Sciences and Engineering Research Council of Canada (NSERC) grant awarded to GP. D.M. was supported by a Canadian Graduate Scholarship from CIHR; A.M.H. was supported by a Vanier Canada Graduate Scholarship from CIHR; M.Y. and D.T. thank the partial financial support from the National Institute of Health, USA (R21AI140418).

We would like to thank the Donnelly Centre (Toronto, Ontario, Canada) and Plasmidsaurus (Eugene, Oregon, USA) for sequencing services.

S.T.C. and D.M. conceived the idea and design of the research; D.M. and A.M.H. designed and ran the BarSeq experiment; D.M. analyzed chemogenetic data, cloned *trxB* and *gor*, made deletion and CRISPRi mutants, ran the *in vitro* assays, and wrote the manuscript. N.L.C., A.M., and D.M. purified TrxB and GOR. D.M. and B.M. performed the ROS detection assays and analyzed the data. D.M. and S.I. performed CRISPRi susceptibility experiments. M.Y. designed MS-40 and MS-40S. D.T. synthesized MS-40 and S.L. synthesized MS-40S. S.T.C. and G.P. edited the manuscript and, together with M.Y. and G.P., supervised the work. All authors contributed to editing and approving the final version of the manuscript.

## AUTHOR AFFILIATIONS

[1]Department of Microbiology, University of Manitoba, Winnipeg, Canada
[2]Department of Chemistry, University of Massachusetts, Lowell, Massachusetts, USA
[3]Department of Medical Microbiology & Infectious Disease, University of Manitoba, Winnipeg, Canada

## AUTHOR ORCIDs

Andrew M. Hogan  http://orcid.org/0000-0002-0782-012X
Silvia T. Cardona  http://orcid.org/0000-0002-0629-8886

## FUNDING

| Funder | Grant(s) | Author(s) |
|---|---|---|
| Canadian HIV Trials Network, Canadian Institutes of Health Research (CTN, CIHR) | | Silvia T. Cardona |
| Gouvernement du Canada \| Natural Sciences and Engineering Research Council of Canada (NSERC) | | Gerd Prehna |
| | | Silvia T. Cardona |
| Canadian HIV Trials Network, Canadian Institutes of Health Research (CTN, CIHR) | | Dustin T. Maydaniuk |
| Canadian HIV Trials Network, Canadian Institutes of Health Research (CTN, CIHR) | | Andrew M. Hogan |
| Foundation for the National Institutes of Health (FNIH) | R21AI140418 | Mingdi Yan |

## AUTHOR CONTRIBUTIONS

Dustin T. Maydaniuk, Conceptualization, Data curation, Formal analysis, Investigation, Methodology, Project administration, Validation, Visualization, Writing – original draft, Writing – review and editing | Brielle Martens, Data curation, Methodology, Validation, Visualization, Writing – review and editing | Sarah Iqbal, Data curation, Methodology, Validation, Visualization | Andrew M. Hogan, Conceptualization, Data curation, Formal analysis, Investigation, Methodology, Validation, Visualization, Writing – review and editing | Neil Lorente Cobo, Methodology, Validation | Anna Motnenko, Methodology, Supervision | Dang Truong, Methodology, Validation, Writing – review and editing | Sajani H. Liyanage, Conceptualization, Funding acquisition, Investigation, Methodology, Project administration, Supervision, Writing – review and editing | Mingdi Yan, Conceptualization, Formal analysis, Funding acquisition, Investigation, Methodology, Project administration, Supervision, Writing – review and editing | Gerd Prehna, Conceptualization, Formal analysis, Funding acquisition, Investigation, Methodology, Project administration, Resources, Supervision, Validation, Visualization, Writing – review and editing | Silvia T. Cardona, Conceptualization, Formal analysis, Funding acquisition, Investigation, Methodology, Project administration, Resources, Supervision, Validation, Visualization, Writing – review and editing

## DATA AVAILABILITY

Raw sequencing data are available from the Sequencing Read Archive (SRA) from NCBI under the BioProject ID PRJNA955738. Supplemental Data File 1 contains gene fitness scores of MS-40 and MS-40S exposures. Supplemental data File 2 contains gene fitness scores of ROS of other ROS inducer exposures.

## ADDITIONAL FILES

The following material is available online.

### Supplemental Material

**File S1 (Spectrum03201-23-S0001.xlsx).** Gene fitness scores of MS-40 and MS-40S exposures.
**File S2 (Spectrum03201-23-S0002.xlsx).** Gene fitness scores of ROS of other ROS inducer exposures.
**Supplemental Tables and Figures (Spectrum03201-23-S0003.docx).** Tables S1-S4; Figures S1-S8.

## Open Peer Review

**PEER REVIEW HISTORY (review-history.pdf).** An accounting of the reviewer comments and feedback.

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
