## [Reviewer comments · Microbiology Spectrum]

Microbiology Spectrum

The mechanism of action of auranofin analogs in *B. cenocepacia* revealed by chemogenomic profiling

Dustin Maydaniuk, Brielle Martens, Sarah Iqbal, Andrew Hogan, Neil Lorente Cobo, Anna Motnenko, Dang Truong, Sajani Liyanage, Mingdi Yan, Gerd Prehna, and Silvia Cardona

Corresponding Author(s): Silvia Cardona, University of Manitoba

Review Timeline:

Submission Date:	September 11, 2023
Editorial Decision:	October 11, 2023
Revision Received:	November 13, 2023
Accepted:	December 6, 2023

Editor: Krisztina Papp-Wallace

Reviewer(s): The reviewers have opted to remain anonymous.

Transaction Report:

DOI: <https://doi.org/10.1128/spectrum.03201-23>

October 11, 2023

Dr. Silvia T. Cardona
University of Manitoba
Microbiology
213 Buller Building
45 Chancellor's Circle
Winnipeg, MB R3T 2N2
Canada

Re: Spectrum03201-23 (The mechanism of action of auranofin analogs in *B. cenocepacia* revealed by chemogenomic profiling)

Dear Dr. Silvia T. Cardona:

Link Not Available

Sincerely,

Krisztina Papp-Wallace

Journals Department
Reviewer comments:

Reviewer #1 (Comments for the Author):

This manuscript reports a thorough study into a repurposed novel antimicrobial, auranofin, and two of its analogues, with the aim of understanding their mechanism of action in Burkholderia species and why the analogues but not auranofin are bactericidal. Specific hypotheses are followed up, such as the idea that the analogues may inhibit the second antioxidant system of gram negative bacteria, or may have better cell penetrance, though much of the data suggest a more complex picture. An impressive breadth of biochemical, microbiological and functional genomic experiments are performed. Overall I found the paper to be well-written and most of the individual experiments are technically sound, but there are some places where I do not understand the

interpretation or important experimental features have not been validated and these need to be addressed before publication.

1. Multiple CRISPR knockdowns are used in this study. The efficiency of the knockdown is not shown. Some of the results with these CRISPRi knockdowns are confusing and suggest a low efficiency, for example the fact that the *rodA* knockdown grew similarly to the dCas9 control (Fig 1C) suggests the reduction may not be very much. The degree to which the CRISPRi actually inhibited expression needs to be addressed for the experiments reporting knockdowns.
2. A related point - assays showing inhibition of TrxB protein in vitro seem robust. The CRISPRi knockdown of *trxB* less so. Is it expected that inhibiting expression of an antibiotic's target will result in increased susceptibility? To me the demonstration of direct inhibition in vitro is sufficient and the knockdown result (where I am unclear on the efficiency, or the expected phenotype if *trxB* is knocked down) confuses things.
3. I am not convinced the 18h stop-point OD drawn from MIC tests in Fig 2B is the correct measure to support the conclusions drawn. The permeability assays provide some evidence that the CRISPR knockdowns are working as expected, but there is not a direct relationship between permeability and the efficacy of auranofin (though this does kind of correlate with the analogues). I suggest further demonstrating the indicated growth/resistance differences using growth curves for selected knockdown/concentration combinations of interest. If the question is about uptake then could the authors use mass spec-based accumulation assays to determine how much of each compound is internalised in the bacteria?
4. The manuscript as it stands is very long and some sections read more like methods - I suggest editing down for clarity.
5. The BarSeq results seem robust and comprehensively analysed, but please add a supplementary table showing quality stats and recovery of mutants after the experiment.
6. Lines 326-329 this is indeed an unexpected result. I do not think that the claim that disrupting these specific genes decreases ROS can be made just from a BarSeq experiment as there may be other effects (independent of ROS) that could influence the final abundance of these mutants. I suggest either adding a follow up experiment (eg. CRISPR knockdown/mutant + ROS-sensitive dye), or qualifying this statement.

Reviewer #2 (Comments for the Author):

In the manuscript Spectrum03201-23, the authors elegantly report the mechanism(s) of bactericidal activity of auranofin analogs against *B. cenocepacia*. The study is of potential importance and experiments are well-documented with easy-to-follow descriptions. Though the authors identified the major target of MS-40 and MS-40S, the BarSeq results detailing the after-effects of exposure to these compounds, which the authors call "mechanisms of action", is largely descriptive - requiring further follow-up work. I have some major comments which the authors should be able to address. Additionally, the authors should carefully fix the Fig. legend/citation issues.

Major comments:

Lines 395-398: This hypothesis about how *gor* mutant has positive fitness is unclear. Additionally, ref# 63 is for *E. coli*. If there is redundancy to *gor* function, how does that effect thioredoxin system? Can the exposure to these compounds activate such alternate pathways?

Line 124: Citation/data should be provided for the 'essential' statement. Is TrxB essential? Do the authors have a TrxB mutant that is resistant to these compounds? They can perform a BarSeq in that mutant background to identify alternate targets. If auranofin is inactive due to low permeability, shouldn't BarSeq also identify some of the permeability genes with positive scores for MS-40 and MS-40S? The authors may want to comment on that.

Is it possible that 8 hours is a big window for the BarSeq fitness calculations, because some mutants can become dominant within the population and authors are missing out on some hits? Maybe, a shorter exposure would allow the authors to fish out early events or other possible mechanisms of action and thus know more about mechanisms of resistance?

Minor comments:

Figure 1: The Legend only has A & B which are for Fig 1B & C.

Line 31: Define GOR in Line 26.

Lines 105-115: Should be moved to Methods.

Lines 120-122 ("For this assay..."): Should be moved to Methods.

Lines 123-124: The correct citation is Fig. 1B.

Lines 136-139: Should be moved to Methods.

Line 148: Fig. 8A?

Staff Comments:

Preparing Revision Guidelines

To submit your modified manuscript, log onto the eJP submission site at <https://spectrum.msubmit.net/cgi-bin/main.plex>. Go to Author Tasks and click the appropriate manuscript title to begin the revision process. The information that you entered when you first submitted the paper will be displayed. Please update the information as necessary. Here are a few examples of required

updates that authors must address:

Please return the manuscript within 60 days; if you cannot complete the modification within this time period, please contact me. If you do not wish to modify the manuscript and prefer to submit it to another journal, please notify me of your decision immediately so that the manuscript may be formally withdrawn from consideration by Microbiology Spectrum.

Reviewer comments:

Note: line numbers correspond to marked-up manuscript

Reviewer #1 (Comments for the Author):

This manuscript reports a thorough study into a repurposed novel antimicrobial, auranofin, and two of its analogues, with the aim of understanding their mechanism of action in Burkholderia species and why the analogues but not auranofin are bactericidal. Specific hypotheses are followed up, such as the idea that the analogues may inhibit the second antioxidant system of gram negative bacteria, or may have better cell penetrance, though much of the data suggest a more complex picture. An impressive breadth of biochemical, microbiological and functional genomic experiments are performed. Overall I found the paper to be well-written and most of the individual experiments are technically sound, but there are some places where I do not understand the interpretation or important experimental features have not been validated and these need to be addressed before publication.

1. Multiple CRISPR knockdowns are used in this study. The efficiency of the knockdown is not shown. Some of the results with these CRISPRi knockdowns are confusing and suggest a low efficiency, for example the fact that the rodA knockdown grew similarly to the dCas9 control (Fig 1C) suggests the reduction may not be very much. The degree to which the CRISPRi actually inhibited expression needs to be addressed for the experiments reporting knockdowns.

R:

Thank you for your comment. Indeed, CRISPRi mutants are knockdowns and the degree of gene silencing when dCas9 is expressed can be variable. However, we have observed effective gene silencing in 100% of the CRISPRi mutants we have created and analysed in our lab (Hogan et al., 2023 Nat. Comm, Supplementary Table 2). We agree Fig. 1C was confusing as described. The similar colors shown for the rodA and dCas9 control do not refer to growth but to percent growth relative to the growth of each individual mutant in the absence of the compounds. That is why the heat map shows the same blue color in the row that correspond to no compound. To clarify this aspect, we have added “relative to no treatment” in Fig 1C legend. Please see **line 1314**

2. A related point - assays showing inhibition of TrxB protein in vitro seem robust. The CRISPRi knockdown of trxB less so. Is it expected that inhibiting expression of an antibiotic's target will result in increased susceptibility? To me the demonstration of direct inhibition in vitro is sufficient and the knockdown result (where I am unclear on the efficiency, or the expected phenotype if trxB is knocked down) confuses things.

R:

We apologize for the oversight. While the aforementioned effect is not observable for every target and every drug, it is common to observe increased susceptibility to a drug in a strain where the target copy number is reduced, compared to the wild type. In this case, the enhanced susceptibility effect of reducing *trxB* expression by CRISPRi could be observed for MS-40 and MS-40S while for auranofin was less evident. As *TrxB* is the target of MS-40, MS-40S and auranofin, the difference was suggestive of auranofin not having an effect due to lack of uptake or efflux. We have clarified this aspect in the text (see **lines 129-132**) before referring to the experiment. We have also added a reference (23) where the enhanced susceptibility phenotype as in vivo evidence of drug-target interaction is explained.

3. I am not convinced the 18h stop-point OD drawn from MIC tests in Fig 2B is the correct measure to support the conclusions drawn. The permeability assays provide some evidence that the CRISPR knockdowns are working as expected, but there is not a direct relationship between permeability and the efficacy of auranofin (though this does kind of correlate with the analogues). I suggest further demonstrating the indicated growth/resistance differences using growth curves for selected knockdown/concentration combinations of interest. If the question is about uptake then could the authors use mass spec-based accumulation assays to determine how much of each compound is internalised in the bacteria?

R:

We thank reviewer 1 for their comment. We are carefully considering the reviewer's suggestions to further explore the data gathered in Fig 2B. We have decided we are unable at this point to make conclusions on the permeability of auranofin based on the mutants we have analyzed. Thus, we have decided to remove figures 2B and 2C from the manuscript. The reviewers' suggestions will be taken into account as we address this aspect outside this work. We did keep figure 2A, which was added to Fig. 1 as 1D.

4. The manuscript as it stands is very long and some sections read more like methods - I suggest editing down for clarity.

R:

Thank you for your comment we have reduced the length by removing paragraphs that were redundant with the methodology (*trxB* and *gor* purification **lines 124-128 and 214-221**) Also, by removing a paragraph on auranofin permeability determinants and two panels of previous figure B we have considerably shortened the manuscript from 47 to 39 pages.

5. The BarSeq results seem robust and comprehensively analysed, but please add a supplementary table showing quality stats and recovery of mutants after the experiment.

R: We have added supplementary table Table S4. with the correlation coefficients across replicates See **447**

and supplemental material. The recovery of mutants after the experiment is shown as gene fitness in supplementary data file 1.

6. Lines 326-329 this is indeed an unexpected result. I do not think that the claim that disrupting these specific genes decreases ROS can be made just from a BarSeq experiment as there may be other effects (independent of ROS) that could influence the final abundance of these mutants. I suggest either adding a follow up experiment (eg. CRISPR knockdown/mutant + ROS-sensitive dye), or qualifying this statement.

R:

We have modified the paragraph accordingly. We mentioned the unexpected observation, but we do not make any conclusions based on just the BarSeq experiment. We have moved figure 6 to the supplementary material. Please see **lines 447-457**.

Reviewer #2 (Comments for the Author):

In the manuscript Spectrum03201-23, the authors elegantly report the mechanism(s) of bactericidal activity of auranofin analogs against *B. cenocepacia*. The study is of potential importance and experiments are well-documented with easy-to-follow descriptions. Though the authors identified the major target of MS-40 and MS-40S, the BarSeq results detailing the after-effects of exposure to these compounds, which the authors call "mechanisms of action", is largely descriptive - requiring further follow-up work. I have some major comments which the authors should be able to address. Additionally, the authors should carefully fix the Fig. legend/citation issues.

Major comments:

- 1) Lines 395-398: This hypothesis about how *gor* mutant has positive fitness is unclear. Additionally, ref# 63 is for *E. coli*. If there is redundancy to *gor* function, how does that affect thioresoxin system? Can the exposure to these compounds activate such alternate pathways?

R: We have reviewed this part of the discussion and modified it accordingly. Indeed, the connexions between the thioresoxin system and GOR function are poorly understood, even in *E. coli*. We suspect there is a redundancy of functions, which may make it difficult to address the function of the individual genetic elements. Please see **lines 726-732** for the new added paragraph

- 2) Line 124: Citation/data should be provided for the 'essential' statement. Is TrxB essential? Do the authors have a TrxB mutant that is resistant to these compounds? They

can perform a Barseq in that mutant background to identify alternate targets.

R:

Yes, TrxB is essential in *B. cenocepacia* K56-2 based on our previous TnSeq library and the new BarSeq library created for this work. We have added a paragraph clarifying the essentiality status of *trxB* and *gor* and added the corresponding reference. Please see **lines 113-115**.

3) If auranofin is inactive due to low permeability, shouldn't BarSeq also identify some of the permeability genes with positive scores for MS-40 and MS-40S? The authors may want to comment on that.

R:

While the idea of auranofin being inactive due to the restrictions of the cell envelope is a likely hypothesis, we have decided to further investigate it in a separate body of research. We did observe some transporter genes with positive fitness scores, some are: CorA (K562_RS09720) with fitness scores of 2.35 and 2.78 for MS-40 and MS-40S, respectively, and a MFS transporter (K562_RS04215) with a fitness score of 1.61 and 1.60 for MS-40 and MS-40S, respectively. In the future, the BarSeq experiment can be repeated with auranofin looking for susceptible mutants in cell envelope genetic determinants.

4) Is it possible that 8 hours is a big window for the Barseq fitness calculations, because some mutants can become dominant within the population and authors are missing out on some hits? Maybe, a shorter exposure would allow the authors to fish out early events or other possible mechanisms of action and thus know more about mechanisms of resistance?

R: We chose 8 hours, which corresponds to 10-12 generations, early stationary phase. Our choice was based on other published work (Geisinger et al. 2020, doi 10.1038/s41467-020-18301-2; Gallagher et al 2011, doi 10.1128/mBio.00315-10; Coe et al 2019; doi 10.1371/journal.ppat.1007862).

While shorter times may allow for detection of more subtle phenotypes, in the conditions we used there were no cases where a single (or even several) mutant dominated the population. We have clarified this aspect in **line 326**.

Minor comments:

Figure 1: The Legend only has A & B which are for Fig 1B & C.

Fixed. See **lines 115, 127, and 137**

Line 31: Define GOR in Line 26.

Fixed. See **line 26**.

Lines 105-115: Should be moved to Methods.

removed and moved to the methods section. See **lines 124-128**

Lines 120-122 ("For this assay..."): Should be moved to Methods. Lines 123-124: The correct citation is Fig. 1B.

removed and moved to the methods section. See **lines 124-128**. Figure number fixed.

Lines 136-139: Should be moved to Methods. Line 148: Fig. 8A?

removed and moved to the methods section. See **lines 214-221** Fig. number fixed (see **line 217**).

Re: Spectrum03201-23R1 (The mechanism of action of auranofin analogs in *B. cenocepacia* revealed by chemogenomic profiling)

Dear Dr. Silvia T. Cardona:

Your manuscript has been accepted, and I am forwarding it to the ASM production staff for publication. Your paper will first be checked to make sure all elements meet the technical requirements. ASM staff will contact you if anything needs to be revised before copyediting and production can begin. Otherwise, you will be notified when your proofs are ready to be viewed.

Sincerely,
Krisztina Papp-Wallace
Editor
Microbiology Spectrum

Reviewer #1 (Comments for the Author):

The revised manuscript is significantly improved and I appreciate the authors' changes. I have the following minor comments.

1. The clarification in the legend to Figure 1 is appreciated. Since this is reporting ratio/percentage data, please add a supplementary dataset with the raw OD values from which Fig1C relative growth data were calculated. Please also add similar data for Fig 5C and make it clear in the legend what the % is relative to (I assume same strain without antibiotic).
2. The response to my previous point 5 with correlation coefficients is helpful but this was not exactly what I meant. I was looking for a supplementary table showing # reads, # mapped and # unique transposon insertions identified in each condition and replicate as this information is useful for readers to understand how comprehensive a dataset is likely to be.

Reviewer #2 (Comments for the Author):

The authors have sufficiently addressed my comments.